# Parainfluenza virus infections in patients with hematological malignancies or stem cell transplantation: Analysis of clinical characteristics, nosocomial transmission and viral shedding

Julia Tabatabai[1,2]*, Paul Schnitzler[1], Christiane Prifert[3], Martin Schiller[4,5], Benedikt Weissbrich[3], Marie von Lilienfeld-Toal[6,7], Daniel Teschner[8], Karin Jordan[4], Carsten Müller-Tidow[4], Gerlinde Egerer[4], Nicola Giesen[4]

1 Department of Infectious Diseases, Virology, University Hospital Heidelberg, Heidelberg, Germany,
2 Center for Child and Adolescent Medicine, University Hospital Heidelberg, Heidelberg, Germany,
3 Institute of Virology and Immunobiology, University Hospital Wuerzburg, Wuerzburg, Germany,
4 Department of Internal Medicine V, University Hospital Heidelberg, Heidelberg, Germany, 5 Department of Internal Medicine, HochFranken Hospitals, Munchberg, Germany, 6 Department of Internal Medicine II, University Hospital Jena, Jena, Germany, 7 Leibniz Institute for Natural Product Research and Infection Biology, Hans-Knöll Institut, Jena, Germany, 8 Department of Hematology, Medical Oncology, & Pneumology, University Medical Center of the Johannes Gutenberg University, Mainz, Germany

* julia.tabatabai@med.uni-heidelberg.de

## Abstract

To assess morbidity and mortality of parainfluenza virus (PIV) infections in immunocompromised patients, we analysed PIV infections in a hematology and stem cell transplantation (SCT) unit over the course of three years. Isolated PIV strains were characterized by sequence analysis and nosocomial transmission was assessed including phylogenetic analysis of viral strains. 109 cases of PIV infection were identified, 75 in the setting of SCT. PIV type 3 (n = 68) was the most frequent subtype. PIV lower respiratory tract infection (LRTI) was observed in 47 patients (43%) with a mortality of 19%. Severe leukopenia, prior steroid therapy and presence of co-infections were significant risk factors for development of PIV-LRTI in multivariate analysis. Prolonged viral shedding was frequently observed with a median duration of 14 days and up to 79 days, especially in patients after allogeneic SCT and with LRTI. Nosocomial transmission occurred in 47 patients. Phylogenetic analysis of isolated PIV strains and combination with clinical data enabled the identification of seven separate clusters of nosocomial transmission. In conclusion, we observed significant morbidity and mortality of PIV infection in hematology and transplant patients. The clinical impact of co-infections, the possibility of long-term viral shedding and frequent nosocomial transmission should be taken into account when designing infection control strategies.

**Data Availability Statement:** Nucleotide sequences retrieved in this study were deposited in

GenBank (accession numbers MT489396-MT489461); this includes the relevant virological data. All relevant clinical data is summarized within the manuscript. However the raw data table for clinical and demographic details of the patient cannot be published due to ethical considerations as the information about age, time of hospital stay and underlying malignancy could be used to de-identify patients. The ethical board in Heidelberg did not allow to publish raw clinical data sets. For data requests please contact the ethical research board Heidelberg, Alte Glockengießerei 11/1, 69115 Heidelberg/Germany, phone: 004962215626460, Mail: ethikkomission-l@med.uni-heidelberg.de.

**Funding:** The author(s) received no specific funding for this work.

**Competing interests:** NG: honoraria from MSD and Roche, advisory board from Pfizer, and travel grants from BMS and Karyopharm; DT – honoraria and travel grants from Gilead, IQone, MSD, and Pfizer, and travel grants from Abbvie, Astellas, Celgene, Jazz; CMT – grants and/or provision of investigational medicinal product from Pfizer, Daiichi Sankyo, and BiolineRx, research support by Deutsche Forschungsgemeinschaft DFG, Deutsche Krebshilfe, BMBF, Wilhelm Sander Stiftung, José Carreras Stiftung, and Bayer AG, and advisory boards from Pfizer, and Janssen-Cilag GmbH. This does not alter our adherence to PLOS ONE policies on sharing data and materials

## Introduction

Respiratory viruses such as influenza, parainfluenza (PIV), respiratory syncytial virus (RSV) and most recently the novel coronavirus SARS-CoV-2 can cause significant morbidity and mortality in immunocompromised patients, in particular in patients with hematologic malignancies or following stem cell transplantation (SCT) [1–3]. While many efforts have been undertaken in research on influenza, much less is known about PIV infections in immunocompromised patients. Several reports describe PIV as a relevant pathogen for immunocompromised patients with mortality rates of PIV-associated lower respiratory tract infection (LRTI) of up to 27% [4–7]. Moreover, PIV is easily transmitted and known to be highly contagious. In contrast to seasonal influenza, PIV infections occur throughout the year. In hematology wards and transplant units, outbreaks of nosocomial PIV infections have been repeatedly reported [6,8–11].

For patients with hematological malignancies presenting with symptoms of respiratory tract infection, testing for respiratory viruses including influenza, PIV and RSV is highly recommended [12]. In contrast to influenza, no specific antiviral therapy has been established against PIV infections [1] and the impact of ribavirin therapy on the outcome of PIV infections remains unclear [13–15].

PIV belongs to the *Paramyxoviridae* family, which comprise enveloped single-stranded negative-sense RNA viruses and is spread by direct contact and aerosols. Based on genetic and antigenic differences PIV types 1–4 have been described, among which PIV type 1 and 3 are classified as members of the genus of *Rubulavirus*, and PIV type 2 and 4 as members of the genus of *Respirovirus* [16–19]. Their major antigenic spike glycolproteins, hemagglutinin neuraminidase and fusion protein, are encoded by HN and F genes, respectively, and are dominant targets for humoral immunity found in all parainfluenza viruses [16]. Further, the HN protein comprises neuraminidase and hemagglutinin functions, and facilitates membrane fusion with host cells by interaction with the F protein [20,21].

Due to its high antigenic and sequence variability the hemagglutinin neuraminidase gene was established as primary target for phylogenetic analysis and typing of PIV [21–25].

Here, we analyze clinical characteristics of PIV infections and risk factors for severe infection in hematological and SCT patients over the course of three years. We assess the extent of nosocomial transmission by combining clinical and molecular data including phylogenetic analysis of viral strains and report on prolonged viral shedding.

## Materials and methods

### Patient population and clinical data assessment

From July 2013 to June 2016, all documented cases of PIV infection in patients with hematologic malignancies or following SCT treated at our institution, a university hospital and transplant center, were included in this analysis. Diagnosis of PIV is established by polymerase chain reaction (PCR) detection of viral RNA in respiratory materials. Patients with PIV infections are regularly re-screened for presence of PIV RNA to determine duration of viral shedding and steer isolation measures.

In this analysis, clinical characteristics and outcome of infected patients were retrospectively evaluated by review of medical charts. PIV-associated LRTI was assumed in case of clinical symptoms of respiratory tract infection (fever, cough, dyspnea) plus atypical pulmonary infiltrates present on thoracic computed tomography (CT) scan in the setting of PIV infection. Severe LRTI was defined as requiring treatment on the intensive care unit (ICU) or fatal outcome. Severe leukopenia was defined as leukocytes < 1000/μl, hypogammaglobulinemia as

immunoglobulin G < 6g/l, and prior steroid therapy as prednisolone ≥ 20mg/day or equivalent.

Nosocomial transmission based on clinical data was assumed in patients with detection of PIV infection ≥ 7 days after hospital admission based on the upper limit of the typical incubation period. Assignment to a specific cluster of nosocomial transmission was based on the following epidemiological case definition: identical viral sequence plus overlapping in-patient stay with at least one other cluster patient while both positive for PIV.

Duration of viral shedding was calculated from first to last positive PIV test, patients with only one available positive test were excluded for this analysis.

## PCR and phylogenetic analysis

Viral RNA was extracted from respiratory specimens using the QIAamp® viral RNA mini kit (Qiagen, Hilden, Germany) according to the manufacturer's protocol. Reverse transcription, amplification and detection of viral RNA was performed with the RealStar® Parainfluenza real-time RT-PCR kit (altona Diagnostics, Hamburg, Germany) on a LightCycler® 480 instrument II (Roche, Mannheim, Germany) according to the manufacturer's instructions.

Extracted RNA was reverse transcribed using random hexamer primers. Subsequently, PIV HN gene was amplified from cDNA using primers for PIV type 1–4 as previously described or adapted by Villaran et al., Echevarria et al. and Abiko et al. [26–28].

Resulting PCR products with an amplicon length between 430–500 nucleotides were sequenced completely in both directions using Big Dye terminator chemistry version 1.1 on a Prism 3130xl instrument (Applied Biosystems, Darmstadt, Germany). Overlapping sequences were assembled using the SEQMAN II software of the Lasergene package (DNAstar, Madison, USA). Multiple alignments from PIV nucleotide sequences were carried out with the MEGA software version 7 [29]. A phylogenetic tree was generated in MEGA using the maximum-likelihood method and the Tamura-Nei algorithm. Representative reference sequences were obtained from GenBank (http://www.ncbi.nlm.nih.gov) and included in the tree. The statistical significance of the tree topology was assessed by bootstrapping with 1,000 replicates to evaluate confidence estimates. Nucleotide sequences retrieved in this study were deposited in GenBank (accession numbers MT489396-MT489461).

## Statistical analysis

The impact of possible influence factors on morbidity and mortality was analyzed by univariate Chi-square tests. Multivariate logistic regression was performed on a reduced set of variables. Factors that might influence duration of PIV shedding were analyzed by Kruskal-Wallis tests. Multivariate logistic regression was performed regarding the endpoint duration of viral shedding > 14 days. In all analyses, p-values < 0.05 were considered as statistically significant.

This study was approved by the ethics committee of the University of Heidelberg (IRB S-090/2018). Patient records and information were anonymized and de-identified prior to analysis, therefore explicit consent was waived by the ethical committee.

## Results

### Clinical characteristics, morbidity and mortality

We identified 109 patients with documented PIV infection between July 2013 and June 2016 (Table 1). The majority of cases was detected during the respective winter and spring seasons (Fig 1). Median age of patients was 60 years [range 26–79], 63% were male. In total 75 patients (69%) had received a SCT (41 allogeneic, 39 autologous, 5 both). Information on PIV subtype

**Table 1. Clinical characteristics.**

|  | Patients with PIV infections N = 109 (100%) |
|---|---|
| **PIV type** |  |
| 1 | 4 (5) |
| 2 | 9 (11) |
| 3 | 68 (79) |
| 4 | 5 (6) |
| Data available: n = 78 |  |
| **Outcome** |  |
| URTI only | 62 (57) |
| LRTI | 47 (43) |
| Severe LRTI | 10 (9) |
| Fatal outcome | 9 (8) |
| **Age median [range]** | 60 years [26–79] |
| **Male sex** | 69 (63) |
| **Underlying malignancy** |  |
| Multiple myeloma | 40 (37) |
| Lymphoma | 20 (18) |
| ALL/LBL | 12 (11) |
| AML/MDS | 30 (28) |
| other | 7 (6) |
| **Uncontrolled malignancy** | 35 (32) |
| **Stem cell transplant recipient** | 75 (69) |
| Allogeneic | 41 (38) |
| Autologous | 39 (36) |
| PIV infection pre-engraftment | 25 (23) |
| **Graft-versus-host-disease** | 21 (19) |
| **Steroid therapy** | 38 (35) |
| **Severe leukopenia** | 50 (46) |
| **Hypogammaglobulinemia** | 57 (67) |
| Data available: n = 85 |  |
| **Co-infections** | 28 (26) |
| **Nosocomial infection** | 47 (43) |

Abbreviations: PIV–parainfluenza virus; URTI–upper respiratory tract infection; LRTI–lower respiratory tract infection; ALL–acute lymphoblastic leukemia; LBL–lymphoblastic lymphoma; AML–acute myeloid leukemia; MDS–myelodysplastic syndrome.

was available in 86 cases showing a vast majority of PIV subtype 3 (n = 68; 79%) followed by subtype 2 (n = 9; 10%), 4 (n = 5; 6%), and 1 (n = 4; 5%).

Any co-infections were detected in 28 patients (26%), co-infections in respiratory specimens in 11 (10%). Most notable were bacterial co-infections detected in blood cultures (n = 9) or respiratory materials (n = 2), fungal co-infections with aspergillus (n = 3), and co-infections with respiratory viruses (1 FLU-B, 2 RSV, 2 coronavirus).

Regarding outcome, 62 patients (57%) had upper respiratory tract infections (URTI) only, 47 patients (43%) developed a LRTI. A severe LRTI was present in 10 patients. 9/47 patients with LRTI died, resulting in a mortality rate of 19%; 1 patient was put on extracorporeal membrane oxygenation (ECMO) and subsequently recovered. Details on fatal cases are given in Table 2. Within 90 days after PIV infection, 4/62 patients (6%) with PIV-URTI as well as 1 patient with PIV-LRTI who since had recovered from the infection died of unrelated causes.

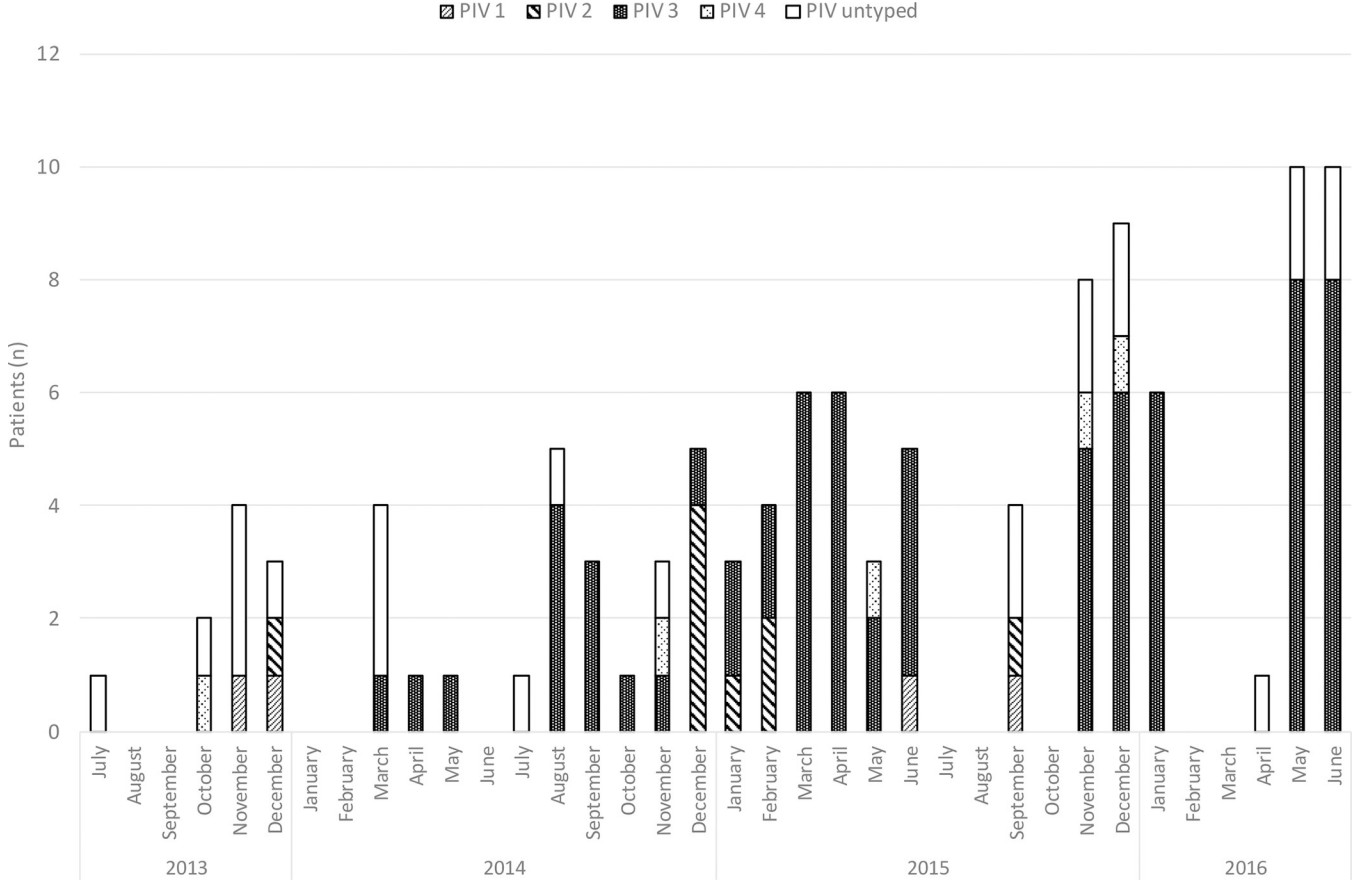

**Fig 1. Timeline of parainfluenza virus infections.** Untyped PIV: Samples were PCR positive, but could not be sequenced for further typing due to low viral loads.

**Table 2. Details on cases of fatal parainfluenza virus infection.**

| # | PIV type | age, years | sex | Underlying malignancy | transplant | Atypical LRTI | Co-infections | Presumed cause of death |
|---|---|---|---|---|---|---|---|---|
| 1 | 2 | 57.1 | M | myeloma | auto-allo | yes | *K. pneumoniae* (BAL), CMV (BAL), *E. coli* (U), *S. epidermidis* (BC) | Septic shock, multi-organ failure |
| 2 | 1 | 73.1 | F | myeloma | - | yes | - | Respiratory failure |
| 3 | untyped | 69.0 | M | PMF | allogeneic | yes | - | ARDS |
| 4 | untyped | 53.0 | M | CLL | allogeneic | yes | - | Respiratory failure |
| 5 | 3 | 65.1 | F | myeloma | autologous | yes | Aspergillus (BAL) | Respiratory failure |
| 6 | 3 | 60.8 | F | FL | autologous | yes | Aspergillus (BAL) | Respiratory failure |
| 7 | 3 | 50.2 | F | AML | allogeneic | yes | - | Cerebral bleeding |
| 8 | untyped | 78.8 | M | DLBCL | - | yes | - | Respiratory failure |
| 9 | 3 | 62.9 | F | myeloma | autologous | yes | - | Respiratory failure |

Abbreviations: PIV–parainfluenza virus; LRTI–lower respiratory tract infection; M–male; F–female; PMF–primary myelofibrosis; CLL–chronic lymphocytic leukemia; FL–follicular lymphoma; AML–acute myeloid leukemia; DLBCL–diffuse large b-cell lymphoma; CMV–cytomegalovirus; BAL–bronchoalveolar lavage; U–urine; BC–blood culture; ARDS–acute respiratory distress syndrome.

## Risk factor analysis regarding morbidity and mortality

Neither type of PIV or underlying hematologic disease had a significant impact on outcome. In particular, no significant association was seen between PIV type 1–4 and development of LRTI (p = 0.81). No increased risk of LRTI, severe LRTI or fatal outcome was seen in patients with prior autologous or allogeneic SCT, even if restricting analysis to patients with SCT within 100 days of PIV diagnosis. Severe leukopenia (p = 0.004), uncontrolled malignancy (p = 0.004), prior steroid therapy (p<0.001), presence of co-infections (p<0.001), and nosocomial transmission (p<0.001) were significantly associated with an increased risk of developing PIV-related LRTI in univariate analysis. In multivariate analysis, severe leukopenia (p = 0.01), prior steroid therapy (p = 0.001), and presence of co-infections (p = 0.01) remained significant risk factors for development of LRTI (Table 3).

With respect to fatal outcome, presence of respiratory tract co-infections (p = 0.02) and prior steroid therapy (p<0.001) showed a significant impact (p = 0.001) in univariate analysis, a trend was seen for male sex (p = 0.05). No parameters reached statistical significance in multivariate analysis.

Patients with PIV infection pre-engraftment did not show a significantly prolonged time-to-engraftment compared to patients with infection post-engraftment neither in case of allogeneic not autologous transplantation (p = 0.81 and p = 0.63, resp.).

Regarding antiviral therapy, ribavirin is not standard of care for PIV infection at our institution. In this cohort, only one patient with PIV LRTI received ribavirin and survived, making any conclusions as towards its effectiveness speculative.

## Viral shedding

Data on viral shedding was available in 40 patients. Median duration of viral shedding was 14 days (range 3–79 days, Fig 2). In univariate analysis, male sex (p = 0.02), severe leukopenia (p = 0.01), prior steroid therapy (p = 0.03), nosocomial acquisition (p = 0.005), LRTI (p = 0.001) and presence of co-infections (p = 0.04) were significantly more frequently associated with prolonged viral shedding. In multivariate analysis, a trend was seen for prolonged viral shedding in patients with allogeneic transplantation (p = 0.07), presence of LRTI (p = 0.09), and severe leukopenia (p = 0.09) (Table 4). Interestingly, available data from 2 patients who acquired PIV infection prior to engraftment after allogeneic SCT showed remarkably prolonged viral shedding for 57 and 79 days, respectively.

## Phylogenetic analysis and assessment of nosocomial transmission

Nosocomial transmission based on clinical data was apparent in 47 patients (43%). Of these, genetic identification of the PIV strain was possible in 38 patients. Combining information on

**Table 3. Multivariate risk factor analysis regarding development of LRTI.**

| Factor | p-value | HR | 95% CI |
|---|---|---|---|
| Allogeneic SCT | 0.89 | 0.92 | 0.29;2.95 |
| Autologous SCT ≤ 100 days | 0.42 | 0.58 | 0.15;2.22 |
| Steroid therapy | 0.001 | 6.03 | 2.15;16.95 |
| Severe leukopenia | 0.01 | 4.96 | 1.46;16.90 |
| Age ≥ 65 years | 0.39 | 1.67 | 0.52;5.37 |
| Co-infections | 0.01 | 4.04 | 1.32;12.36 |

Abbreviations: LRTI–lower respiratory tract infection; HR–hazard ratio; 95% CI– 95% confidence interval; SCT–stem cell transplantation.

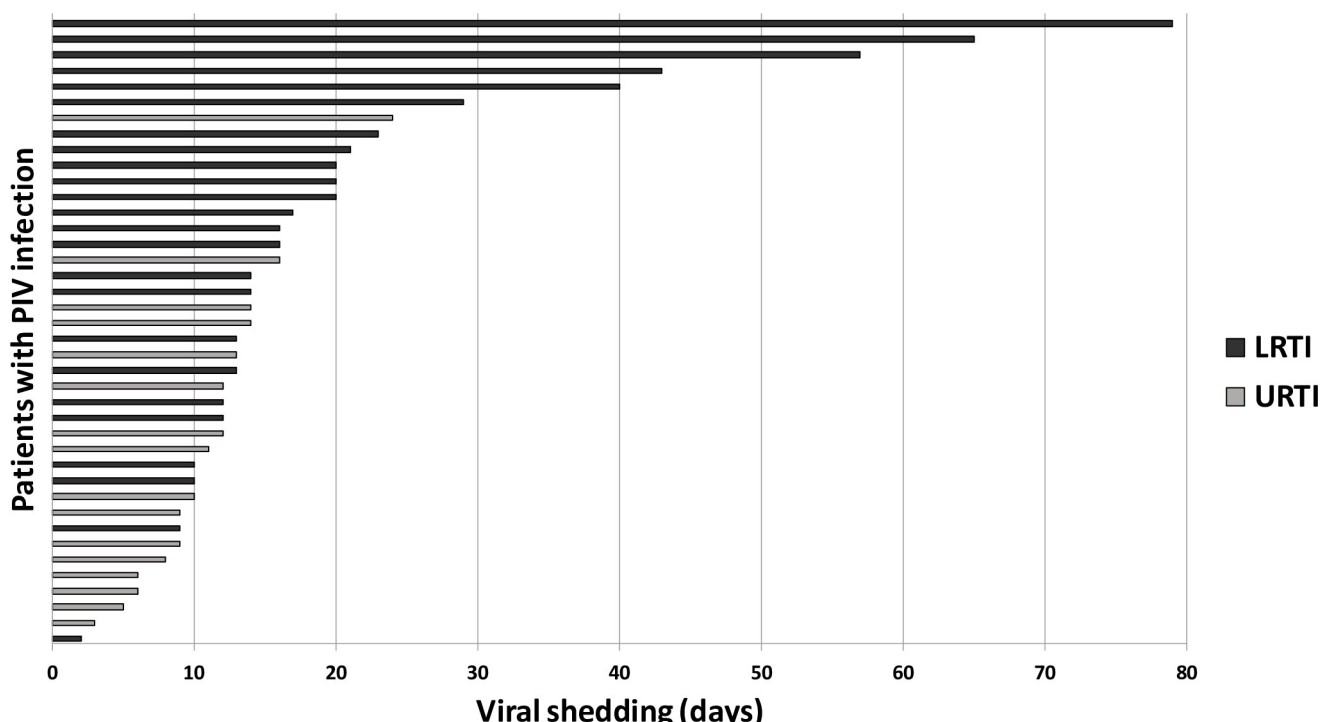

**Fig 2. Duration of viral shedding in patients with PIV infection.** Data on viral shedding was available in 40 patients with consecutive tests for PIV. Patients with URTI and LRTI are designated by green and red bars, resp.

nosocomial transmission according to clinical definition with phylogenetic data on viral strains, we could identify seven clusters of nosocomial PIV infections consisting each of patients with clinically defined nosocomial PIV infection, overlapping stays as in-patients and identical viral sequence. The identified clusters included up to seven patients each and were spread over a period of 23 months (Fig 3). Two nosocomial clusters of three patients each were located within the same phylogenetic cluster but occurred during different time periods (PIV3 C3d, 08-10/14, 04/15). Out of 38 patients with nosocomially acquired PIV infection and available sequence data, 33 patients (87%) could be assigned to one of the clusters. In addition, seven patients with presumably community-acquired PIV infection showed viral sequences identical to one of the clusters, three of these were hospitalized within the PIV incubation period but shorter than the upper limit of standard incubation period and might be in fact nosocomial cases. Furthermore, three patients with community-acquired PIV infection formed an additional cluster (PIV3 C3a1, 06/2016). All three were treated during the presumed time of

**Table 4. Multivariate risk factor analysis regarding prolonged viral shedding > 14 days.**

| Factor | p-value | HR | 95% CI |
|---|---|---|---|
| Allogeneic SCT | 0.07 | 8.63 | 0.84;88.72 |
| Steroid therapy | 0.61 | 1.62 | 0.26;10.12 |
| LRTI | 0.09 | 6.29 | 0.76;52.21 |
| Severe leukopenia | 0.09 | 7.42 | 0.73;74.90 |

Abbreviations: HR–hazard ratio; 95% CI– 95% confidence interval; SCT–stem cell transplantation; LRTI–lower respiratory tract infection.

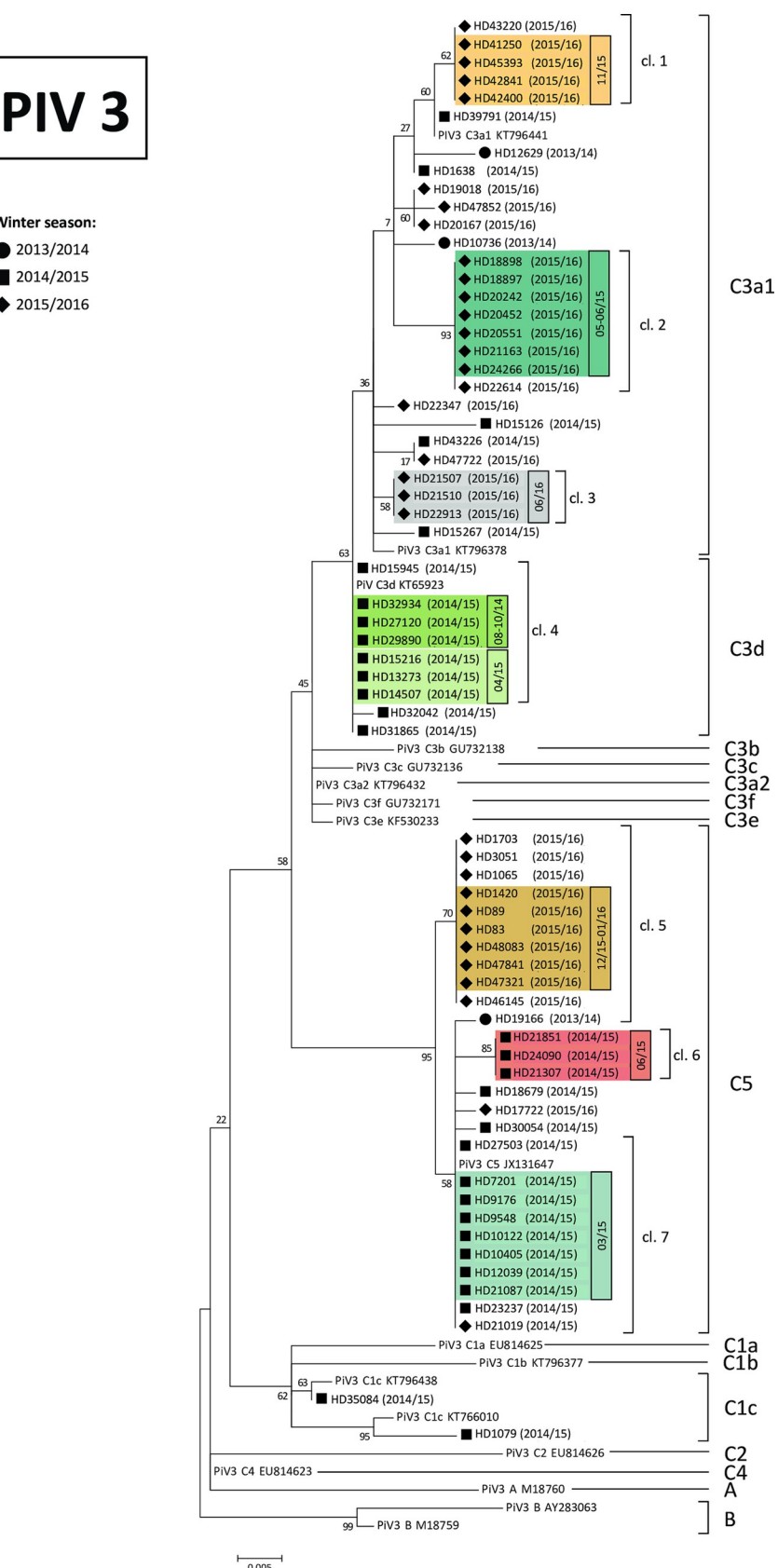

**Fig 3. Phylogenetic analysis of PIV strains including information on clusters of nosocomial transmission.**
Phylogenetic tree for nucleotide sequences of PIV-3 strains were constructed with maximum-likelihood method with 1,000 bootstrap replicates using MEGA 7 software. Heidelberg strains are named with their strain identifier followed by the winter season of isolation in brackets. Reference strains representing known genotypes were retrieved from GenBank and included in the tree (labels include genotype followed by accession number). The genotype assignment is also shown on the right by brackets. Bootstrap values greater than 70% are indicated at the branch nodes. Clinically suspected nosocomial infections matching identical sequence clusters (cl. 1–7) are highlighted in color (one cluster of suspected nosocomial infection in the outpatient setting is highlighted in grey), time of infection is shown in black circled box on the right. The scale bar represents the number of nucleotide substitutions per site. cl. = cluster.

infection in the allogeneic transplant outpatient clinic, thus nosocomial transmission in the waiting area might be conceivable.

## Discussion

This multi-season study of PIV infections in a diverse population of patients with hematologic malignancies including both SCT and non-SCT patients shows significant morbidity and mortality with nearly half of infected patients developing pneumonia and a subsequent LRTI-associated mortality rate of 19%. The incidence of severe courses of PIV infection seen here is within range of those reported by others, taking into consideration that most published studies focused on high-risk populations such as patients with leukemia or following SCT [5,7,14,30,31]. In our study population, SCT status was not a significant risk factor for severe outcome. This highlights the role of PIV as an important pathogen in patients with hematologic malignancies both within and outside the SCT setting.

While PIV type 3 has been associated with an increased incidence of LRTI in hematologic patients [32] we could not detect a significant association between PIV type and development of LRTI. However, in our cohort PIV type 3 was responsible for nearly 4 in 5 of overall PIV infections. Of interest, among the six fatal cases with information on PIV type, two were associated with PIV other than type 3, namely type 1 and 2, respectively.

Prior steroid therapy, severe leukopenia and presence of co-infections were identified as significant risk factors for PIV-LRTI. We observed bacterial, fungal and viral co-infections. Of interest, in five cases co-infections with other respiratory viruses including two cases of co-infection with coronavirus (non-COVID-19) were detected. However, there was no noticeable associated increase in morbidity in these cases. Presence of co-infections has been repeatedly described as a risk factor for severe PIV infection [14,30,33]. Recently, invasive pulmonary aspergillosis (IPA) as a complication of severe influenza has been gaining a lot of attention with reported incidence rates of 30% of immunocompromised ICU patients and high associated mortality [34]. We observed three cases of IPA and PIV co-infection. All three required treatment on the ICU, two subsequently died, one patient recovered following ECMO therapy. This demonstrates the potential severity of IPA in immunocompromised patients with PIV infection. It is therefore important to aim for thorough microbiological work-up in patients with PIV infection, particularly in the immunocompromised host, in order to detect possible co-infections and adapt antimicrobial therapy accordingly.

Therapeutic options targeting PIV are currently very limited. Antiviral therapy with ribavirin is highly controversial with most studies failing to show a significant impact on LRTI development or mortality [15]. Intravenous immunoglobulin administration may be considered as supportive therapy [1]. An antiviral agent currently in phase III development for PIV infection is the sialidase fusion protein fludase (DAS181). First data suggest fludase may be an effective treatment strategy for PIV LRTI in immunocompromised patients [35]. However, until effective antiviral agents are broadly available, infection control measures remain the cornerstone against PIV infections.

To optimize infection control measures, assessment of viral shedding can be a helpful strategy. We could demonstrate prolonged viral shedding of up to 79 days, particularly in patients with LRTI, severe leukopenia, and allogeneic SCT recipients. While too few to gain statistical significance, PIV infection pre-engraftment of allogeneic SCT seemed a high-risk constellation for prolonged viral shedding. Long-term viral shedding of influenza, PIV, and RSV in immunocompromised patients has been previously reported by our group with especially long periods of nearly a year observed for RSV [36] and has also been described for the novel coronavirus SARS-CoV-2 [37]. The possibility of long-term viral shedding has to be kept in mind when devising infection control strategies as it might facilitate nosocomial transmission and outbreaks.

Clinically suspected nosocomial transmission supported by sequence analysis was a frequent finding in our study cohort despite comprehensive hygienic measures implemented at our institution. This highlights the high contagiousness of PIV, especially in such a vulnerable patient population. Outbreaks of PIV on hematology and oncology wards and in SCT units have been repeatedly reported including both outbreaks of a single and multiple virus strains [6,8,10,11,38,39]. We here describe multiple clusters of nosocomial transmission in immunocompromised patients outside of a traditional outbreak setting covering a long time period. The combination of clinical and phylogenetic data allowed a detailed case-by-case analysis and to illustrate the route and extent of nosocomial transmissions. Clusters of nosocomial transmission could be observed during all four seasons reflecting the presence of PIV throughout the year, highlighting the need to implement adequate infection control measures at any time. Circle threshold values in real time PCR as proxy for viral load did not show any association with LRTI nor severe LRTI in our cohort. However, it is very conceivable that a prolonged period of viral shedding, such as here observed in allogeneic transplant patients increases the risk of nosocomial transmission. At our institution, isolation of not only infected patients but also their contact patients for the length of the possible incubation period is standard of care which might have contributed to stop the development of larger outbreaks despite the obviously repeated introduction of PIV into this highly vulnerable patient population. Barrier methods addressing the entire population at risk such as a universal mask strategy if in contact with SCT patients have also been shown to be effective in reducing PIV infections [40].

As a retrospective analysis, this study has several limitations. Detailed documentation of clinical symptoms as well as stringent follow-up swabs to determine duration of viral shedding were not available in all patients, especially in the out-patient setting. Furthermore, testing for PIV was limited to patients. Thus, no information on PIV infections among health-care workers or patients' relatives was available which would have added useful aspects with regard to chains of transmission.

In conclusion, we could demonstrate significant morbidity and mortality of PIV infections in a diverse population of hematologic and SCT patients. Nosocomial transmission occurred frequently and might be facilitated by long-term viral shedding in immunocompromised patients highlighting the need for comprehensive infection control management. Further prospective studies are necessary to design optimal strategies with regard to infection prevention and transmission control in this vulnerable patient population, and to further develop efficient vaccination and treatment options.

## Author Contributions

**Conceptualization:** Benedikt Weissbrich.

**Data curation:** Christiane Prifert, Karin Jordan, Carsten Müller-Tidow, Gerlinde Egerer, Nicola Giesen.

**Formal analysis:** Christiane Prifert, Nicola Giesen.

**Investigation:** Nicola Giesen.

**Methodology:** Julia Tabatabai, Paul Schnitzler, Christiane Prifert, Benedikt Weissbrich, Daniel Teschner, Nicola Giesen.

**Project administration:** Paul Schnitzler.

**Supervision:** Paul Schnitzler, Gerlinde Egerer.

**Validation:** Julia Tabatabai, Marie von Lilienfeld-Toal.

**Visualization:** Julia Tabatabai, Benedikt Weissbrich.

**Writing – original draft:** Julia Tabatabai, Nicola Giesen.

**Writing – review & editing:** Julia Tabatabai, Paul Schnitzler, Christiane Prifert, Martin Schiller, Benedikt Weissbrich, Marie von Lilienfeld-Toal, Daniel Teschner, Karin Jordan, Carsten Müller-Tidow, Gerlinde Egerer, Nicola Giesen.

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
