## [Decision Letter · Decision Letter 0]

13 Apr 2022

PONE-D-21-30394Parainfluenza Virus Infections in Patients with Hematological Malignancies or Stem Cell Transplantation: Analysis of Clinical Characteristics, Nosocomial Transmission and Viral SheddingPLOS ONE

Dear Dr. Tabatabai,

Thank you for submitting your manuscript to PLOS ONE. After careful consideration, we feel that it has merit but does not fully meet PLOS ONE’s publication criteria as it currently stands. Therefore, we invite you to submit a revised version of the manuscript that addresses the points raised during the review process.

We look forward to receiving your revised manuscript.

Kind regards,

Ahmed S. Abdel-Moneim, Ph.D.

Academic Editor

PLOS ONE

“I have read the journal's policy and the authors of this manuscript have the following competing interests outside of the submitted work: NG: honoraria from MSD and Roche, advisory board from Pfizer, and travel grants from BMS and Karyopharm; DT – honoraria and travel grants from Gilead, IQone, MSD, and Pfizer, and travel grants from Abbvie, Astellas, Celgene, Jazz; CMT – grants and/or provision of investigational medicinal product from Pfizer, Daiichi Sankyo, and BiolineRx, research support by Deutsche Forschungsgemeinschaft DFG, Deutsche Krebshilfe, BMBF, Wilhelm Sander Stiftung, José Carreras Stiftung, and Bayer AG, and advisory boards from Pfizer, and Janssen-Cilag GmbH. The other authors state no competing interests.”

Reviewers' comments:

Reviewer's Responses to Questions

**Comments to the Author**

1. Is the manuscript technically sound, and do the data support the conclusions?

Reviewer #1: Yes

Reviewer #2: Yes

2. Has the statistical analysis been performed appropriately and rigorously? 

Reviewer #1: Yes

Reviewer #2: Yes

3. Have the authors made all data underlying the findings in their manuscript fully available?

Reviewer #1: Yes

Reviewer #2: Yes

4. Is the manuscript presented in an intelligible fashion and written in standard English?

Reviewer #1: Yes

Reviewer #2: Yes

5. Review Comments to the Author

Reviewer #1: This is an interesting and well written manuscript that throws light on an important virus that caused outbreaks and high mortality among the immunocompromised patients.

My recommendations to authors

Please include the epidemiological case definition used to assign patients to an outbreak or cluster. Please include any detected risk factors that increased transmission, and if these risk factors were addressed.

Reviewer #2: The authors analyzed PIV infections in a hematology and stem cell transplantation (SCT) unit over the course of three years. Isolated PIV strains were characterized by sequence analysis and nosocomial transmission was assessed including phylogenetic analysis of viral strains. This is a complete and important study, which needs some improvement in the phylogenetic analysis and some edition for publication.

Major comments:

1. Page 3, lines 65-66: it is stated that phylogeny is based on the F gene but in this study the authors sequenced the HN gene for phylogenetic analysis, as would be expected. Page 3, lines 61-66: This paragraph is important to sustain part of one the aim of this manuscript and is described somehow very superficially. It should be edited for style improvement. In addition, reference is (are) missing.

2. The length of the amplicon used for phylogenetic analysis is not mentioned in any part of the manuscript.

3. I assume that the typing of PIV according to time and presented in Figure 1 is based on the same sequences used for phylogenetic analysis. Why were some samples untypable? Were they included in the phylogenetic tree? Could the size of the sequence be responsible for a lack of discrimination for these untypable samples?

4. The size of the sequence deposited in GenBank is of 438 bp. Is this size sufficient for a good phylogenetic analysis and to propose nosocomial transmission? In one of the references cited (Abiko et al., 2013), the size of the sequence analyzed is quite larger (1599 bp).

5. Putative nosocomial transmission is presented with colors in the phylogenetic tree. However, other samples exhibit an identical sequence but are not associated with the cluster of putative nosocomial transmission. The authors could perform a blast analysis with their samples to see if there are other samples deposited, with identical sequences in the 438 nt analyzed for the phylogenetic analysis. Thus, in page 9, line 215, the authors should suggest nosocomial transmission, instead of affirming it, unless the previous issues responded.

6. Is a previous study (Lefeuvre C et al., JMV 2021), LRTI was associated with HPIV-3. No mention is done in this study is ther was an association of LRTI with HPIV type. In the fatal cases, two types were detected, in addition to untypable samples. The authors should analyze these findings.

7. No mention is given to the Ct values. Was there any difference in Ct values between URTI and LRTI or during the sheding?

Minor comments

8. Page 5, line 119: the number 75 at the beginning of the sentence should be written in letters.

9. Page 6, line 127: Abbreviation URTI is not defined.

---

## [Author Response · Author response to Decision Letter 0]

10 Jun 2022

Letter of rebuttal / comments on changes in the manuscript

Ms. Ref. No.: PONE-D-21-30394

Title: Parainfluenza Virus Infections in Patients with Hematological Malignancies or Stem Cell Transplantation: Analysis of Clinical Characteristics, Nosocomial Transmission and Viral Shedding

Plos One

Questions and answers to the reviewers:

Reviewer#1: This is an interesting and well written manuscript that throws light on an important virus that caused outbreaks and high mortality among the immunocompromised patients.

My recommendations to authors: Please include the epidemiological case definition used to assign patients to an outbreak or cluster. Please include any detected risk factors that increased transmission, and if these risk factors were addressed.

Response: We thank Reviewer #1 for finding our manuscript interesting and well written. The reviewer raises a very important point in asking to include a clear epidemiological case definition which we added to the methods section of the revised manuscript. Similarly, the question of risk factors for increased transmission is very interesting. Circle threshold values in real time PCR as proxy for viral load did not show any association with LRTI nor severe LRTI in our cohort. However, it is quite conceivable that prolonged viral shedding, such as found in allogeneic transplant patients in our cohort, increases the risk for transmission. In the revised manuscript, we added a paragraph on possible risk factors for transmission to the discussion section.

Reviewer#2: The authors analyzed PIV infections in a hematology and stem cell transplantation (SCT) unit over the course of three years. Isolated PIV strains were characterized by sequence analysis and nosocomial transmission was assessed including phylogenetic analysis of viral strains. This is a complete and important study, which needs some improvement in the phylogenetic analysis and some edition for publication.

Major comments:

1. Page 3, lines 65-66: it is stated that phylogeny is based on the F gene but in this study the authors sequenced the HN gene for phylogenetic analysis, as would be expected. 

Response: We thank Reviewer#2 for carefully reading this manuscript as indeed we have sequenced the HN gene and there was a typing error we have corrected accordingly in the revised manuscript.

Reviewer#2: Page 3, lines 61-66: This paragraph is important to sustain part of one the aim of this manuscript and is described somehow very superficially. It should be edited for style improvement. In addition, reference is (are) missing.

Response: We appreciate the reviewers wish for more detail on the genetic and molecular epidemiology of PIV. We have therefore edited this paragraph extensively and added further references.

Reviewer#2: 2. The length of the amplicon used for phylogenetic analysis is not mentioned in any part of the manuscript.

Response: The length of the amplicon varied depending on the PIV type between 430 and 500 nt, we have added this information to the methods section.

Reviewer#2: 3. I assume that the typing of PIV according to time and presented in Figure 1 is based on the same sequences used for phylogenetic analysis. Why were some samples untypable? Were they included in the phylogenetic tree? Could the size of the sequence be responsible for a lack of discrimination for these untypable samples?

Response: The reviewer is raising an important issue. Due to low viral load (accordingly high ct-values), some samples could not be sequenced and typed, so we had only information from the monoplex PCR (altona diagnostics) stated whether samples were PIV positive or negative. We have added this information to the legend of Figure 1 in the manuscript.

Reviewer#2: 4. The size of the sequence deposited in GenBank is of 438 bp. Is this size sufficient for a good phylogenetic analysis and to propose nosocomial transmission? In one of the references cited (Abiko et al., 2013), the size of the sequence analyzed is quite larger (1599 bp). 

Response: We thank the reviewer for this important question. As we have primarily defined nosocomial infection based on clinical criteria and used phylogenetic data in order to support clusters of nosocomial infection, we believe that the sequence length we have used is suitable for that purpose. In contrast, for extensive molecular epidemiological studies complete sequencing of the HN gene might be ideal. Therefore, we believe that the sequence length used in this study is sufficient for our study design. We have added additional information on the definition of nosocomial infection in the methods section.

Reviewer#2: 5. Putative nosocomial transmission is presented with colors in the phylogenetic tree. However, other samples exhibit an identical sequence but are not associated with the cluster of putative nosocomial transmission. The authors could perform a blast analysis with their samples to see if there are other samples deposited, with identical sequences in the 438 nt analyzed for the phylogenetic analysis. Thus, in page 9, line 215, the authors should suggest nosocomial transmission, instead of affirming it, unless the previous issues responded.

Response: The reviewer is mentioning the presentation of suspected nosocomial infection as marked in the phylogenetic tree. As stated in the Figure legend “clinically suspected nosocomial infections matching identical sequence clusters (cl. 1-7) are highlighted in color”, meaning that we have defined nosocomial infection solely based on clinical criteria and marked it in the phylogenetic tree to see if this was supported by identical sequences. We believe that this information is indeed supporting the clinical suspicion of nosocomial infection, but of course sequence analysis as done here does not finally prove nosocomial infection. We have therefore edited our statement in the discussion section accordingly.

Reviewer#2: 6. Is a previous study (Lefeuvre C et al., JMV 2021), LRTI was associated with HPIV-3. No mention is done in this study is there was an association of LRTI with HPIV type. In the fatal cases, two types were detected, in addition to untypable samples. The authors should analyze these findings.

Response: We thank Reviewer #2 for raising the important issue of PIV subtype and outcome. We did not observe a significant association between PIV type and LRTI incidence and added this statement more prominently to the results section. While no association between PIV type and LRTI was detected, PIV type 3 accounted for nearly 4 in 5 infections overall. Furthermore, as the reviewer mentions, the occurrence of fatal non-PIV3 infections is noticeable. We added a paragraph to discuss these findings, also including the suggested study by Lefeuvre et al, to the discussion section of the revised manuscript.

Reviewer#2: 7. No mention is given to the Ct values. Was there any difference in Ct values between URTI and LRTI or during the shedding?

Response: We thank the reviewer for this interesting question. Circle threshold values in real time PCR as proxy for viral load did not show any association with LRTI nor severe LRTI in our cohort. We have added this information to the manuscript.

Reviewer#2: Minor comments

8. Page 5, line 119: the number 75 at the beginning of the sentence should be written in letters.

9. Page 6, line 127: Abbreviation URTI is not defined.

Response: We thank for the reviewers careful reading of the manuscript and have edited the manuscript accordingly.

---

## [Editor Report · Decision Letter 1]

7 Jul 2022

Parainfluenza Virus Infections in Patients with Hematological Malignancies or Stem Cell Transplantation: Analysis of Clinical Characteristics, Nosocomial Transmission and Viral Shedding

PONE-D-21-30394R1

Dear Dr. Tabatabai,

We’re pleased to inform you that your manuscript has been judged scientifically suitable for publication and will be formally accepted for publication once it meets all outstanding technical requirements.

Kind regards,

Ahmed S. Abdel-Moneim, Ph.D.

Academic Editor

PLOS ONE

---

## [Editor Report · Acceptance letter]

12 Jul 2022

PONE-D-21-30394R1 

Parainfluenza virus infections in patients with hematological malignancies or stem cell transplantation: analysis of clinical characteristics, nosocomial transmission and viral shedding 

Dear Dr. Tabatabai:

I'm pleased to inform you that your manuscript has been deemed suitable for publication in PLOS ONE. Congratulations! Your manuscript is now with our production department. 

Kind regards, 

on behalf of

Prof. Ahmed S. Abdel-Moneim 

Academic Editor

PLOS ONE